# A Structure-Aware Framework for Learning Device Placements on Computation Graphs

**Shukai Duan**[*]
Center for Complex Particle Systems
University of Southern California
Los Angeles, USA
shukaidu@usc.edu

**Heng Ping**[*]
University of Southern California
Los Angeles, USA
hping@usc.edu

**Nikos Kanakaris**[*]
University of Southern California
Los Angeles, USA
kanakari@usc.edu

**Xiongye Xiao**[*]
Center for Complex Particle Systems
University of Southern California
Los Angeles, USA
xiongyex@usc.edu

**Panagiotis Kyriakis**[*]
Meta
pkyriakis@meta.com

**Nesreen K. Ahmed**
Cisco Outshift
nesahmed@cisco.com

**Peiyu Zhang**
University of Southern California
Los Angeles, USA
pzhang65@usc.edu

**Guixiang Ma**
Intel Labs
guixiang.ma@intel.com

**Mihai Capotă**
Intel Labs
mihai.capota@intel.com

**Shahin Nazarian**
University of Southern California
Los Angeles, USA
shahin.nazarian@usc.edu

**Theodore L. Willke**
Intel Labs
ted.willke@intel.com

**Paul Bogdan**
Center for Complex Particle Systems
University of Southern California
Los Angeles, USA
pbogdan@usc.edu

## Abstract

Computation graphs are Directed Acyclic Graphs (DAGs) where the nodes correspond to mathematical operations and are used widely as abstractions in optimizations of neural networks. The device placement problem aims to identify optimal allocations of those nodes to a set of (potentially heterogeneous) devices. Existing approaches rely on two types of architectures known as grouper-placer and encoder-placer, respectively. In this work, we bridge the gap between encoder-placer and grouper-placer techniques and propose a novel framework for the task of device placement, relying on smaller computation graphs extracted from the OpenVINO toolkit. The framework consists of five steps, including graph coarsening, node representation learning and policy optimization. It facilitates end-to-end training and takes into account the DAG nature of the computation graphs. We also propose a model variant, inspired by graph parsing networks and complex network analysis, enabling graph representation learning and jointed, personalized graph partitioning,

---

[*]Equal contribution.

38th Conference on Neural Information Processing Systems (NeurIPS 2024).

using an unspecified number of groups. To train the entire framework, we use reinforcement learning using the execution time of the placement as a reward. We demonstrate the flexibility and effectiveness of our approach through multiple experiments with three benchmark models, namely Inception-V3, ResNet, and BERT. The robustness of the proposed framework is also highlighted through an ablation study. The suggested placements improve the inference speed for the benchmark models by up to $58.2\%$ over CPU execution and by up to $60.24\%$ compared to other commonly used baselines.

# 1   Introduction

The ability of intelligent agent systems (IASs) and cyber-physical systems (CPSs) to perceive and accurately interpret complex environments is crucial for artificial intelligence (AI). Recently, there has been a remarkable progress in machine learning and AI, due to the wide adoption of the transformer architecture and foundation models (FMs) [26, 30]. FMs have allowed both academia and industry to perform several data-demanding tasks, ranging from image and text analysis to multi-modal content generation and human-like visual perception [18, 20, 11, 4]. This is achievable due to the self-supervised nature of the FMs, their ability to easily generalize and the large amounts of data available online [29]. The aforementioned properties make FMs distinguishable in specific general pre-training tasks such as next-word prediction, compared to traditional ML architectures that use supervised learning [2]. This recent surge in using large FMs has led to increased demand for computing power, which is projected to grow even more in the next few years [16]. This is due to the need for more advanced model training processes and the continuous expansion of model parameters [26, 6]. It is clear that as FMs become increasingly complex, they demand vast computational resources not only for training but also for fine-tuning and inference tasks. This surge in computational demand underscores the necessity for managing the available hardware more effectively.

In light of this, the concept of device placement has gained popularity lately as a manner to speed up and improve the inference time of deep learning models, including FMs, in systems with a mixture of heterogeneous devices, such as CPUs, GPUs and NPUs [31, 1, 33, 15, 21]. With neural networks evolving towards larger models, heterogeneous and multi-device computing has played a critical role in their implementation. Device placement emerges as a pivotal factor determining the performance of an implementation of a model. Strategically allocating neural networks across multiple devices can significantly reduce the runtime of a model and the overall energy consumption [9]. The current process of device placement typically involves converting a neural network into a computation graph, where each node corresponds to an operation within the neural network. The computation graph is then partitioned and its nodes are allocated to the appropriate devices for processing. The effectiveness of device placement directly impacts the deployment performance of neural networks.

Early on device placement has been the main responsibility of human experts [21]. Engineers with a substantial level of expertise and diligence were responsible for allocating each part of a model to the best-suited device. However, this rigorous task can be daunting, considering the rapid advancement in hardware, which leads to a serious increase in development time, bug fixing, and code optimization [3, 19, 23]. Deep reinforcement learning (DRL) has recently been proposed to provide effective device placements with full automation [31, 1, 21]. Two different DRL architectures for device placement currently exist in the literature: the 'grouper-placer' model that reduces the action space by merging operations into groups; and the 'encoder-placer' that encodes the features of the operations to capture the topological properties of the computational graph [15, 31].

Although existing approaches have been successful, there is still space for improvement. In particular, they demonstrate several shortcomings. To begin with, they disregard the directed and acyclic nature of computation graphs [22]. Furthermore, they either follow a grouper-placer or an encoder-placer architecture [22, 21]. In addition, most of them are not designed to train all of their components simultaneously in an end-to-end fashion and they fail to capture higher-order interactions among the operations of a computation graph. Finally, they make use of large, fine-grained computation graphs, thereby exhibiting slow convergence and demanding a higher number of iterations during the learning process [15].

Considering the limitations mentioned above, this paper proposes a framework to optimize for device placement based on smaller, coarsened computation graphs produced by the OpenVINO

toolkit. Our framework consists of five steps: First, the neural network model is converted into a computation graph. Then, local and global structural features as well as positional, node-specific and fractal features are extracted to compose the initial node feature vectors. Following that, graph representation learning, graph partitioning and pooling are learned jointly, facilitating the fusion of the grouper-placer and encoder-placer models. Finally, we use the execution time of the suggested device placements as a reward to train the entire framework. In contrast with existing approaches, our framework allows for encoding and grouping operations of a computation graph jointly in an end-to-end fashion. Along with the proposed framework, through one of the variant models, we introduce a novel method tailored to computation graphs, for jointly learning node embeddings and performing personalized graph partitioning with an unspecified number of groups for further coarsening. The effectiveness and robustness of the proposed approach are demonstrated through multiple experiments with different benchmark models and a detailed ablation study.

**Contributions**. The main contributions of this paper are the following:

- To the best of our knowledge, this is the first flexible framework for the task of device placement capable of learning graph and node representations as well as graph partitions and pooling jointly in an end-to-end fashion. Even more, we introduce the concept of learning personalized graph partitions using an unspecified number of groups.
- We propose a structure-aware device placement framework that integrates graph coarsening, node representation learning, policy optimization and effectively combines the strengths of grouper-placer and encoder-placer models.
- Our framework is the first of its kind that encodes features from multifractal analysis, positional encodings, and node-specific features for the task of device placement through a model variant, and discusses the impact of incorporating different properties on the model.
- The proposed variant of the framework achieves a state-of-the-art performance improvement of up to $58.2\%$ over CPU execution and $60.24\%$ in comparison with other baseline models.

## 2 Proposed framework

In this section we introduce our framework titled Hierarchical Structure-Aware Device Assignment Graph (HSDAG). It consists of five steps as shown in Figure 1. Briefly, we first convert a neural network model into a computation graph. Then, we extract features for each node and edge of the computation graph. The next step enriches these features via graph representation learning techniques and simultaneously learns how to partition and pool the graphs. The learned features and groups of nodes are utilized to train a stochastic policy, which we use for assigning each node of a graph to the most appropriate device. We train the entire pipeline end-to-end with the objective of minimizing the inference time of the proposed device placement.

### 2.1 Problem Formulation

**Definition 2.1. (Computation graph)**. We denote a computation graph as $G = (V, E)$. $G$ is labeled, unweighted, directed and acyclic with a set of nodes $V = \{v_1, v_2, ..., v_{|V|}\}$ representing operations and a set of edges $E \subseteq V \times V$ representing their connections. Each graph $G$ is associated with a binary asymmetric adjacency matrix as $A \in \{0, 1\}^{|V| \times |V|}$. Each node $v$ of $G$ represents an operation applied to the input data and is associated with an operation type $t_v \in T$. In the context of this paper, the terms node and operation are used interchangeably. An edge $e = (v, u) \in E$ represents the flow of data or dependency among node $v$ and node $u$.

**Definition 2.2. (Device placement)**. Given a list $\mathcal{D}$ of the available devices, a placement $P = \{p_1, p_2, ..., p_n\}$ assigns each operation $v$ of a computation graph $G$ to a device $p \in \mathcal{D}$, where $p \in \{1, 2, ..., |\mathcal{D}|\}$.

**Problem setup.** We focus on the problem of device placement in a heterogeneous computing system. Our goal is to assign each part of a computation graph to the most suitable device, such that the overall execution time during the inference of the model is minimized. Formally, given a computation graph $G$, we learn a policy $\pi : G \to P$ that assigns a placement $p$ for all $v \in G$ such that

$$r^*_{\pi, P} = \max_{\pi, P} r(G; \pi, P), \tag{1}$$

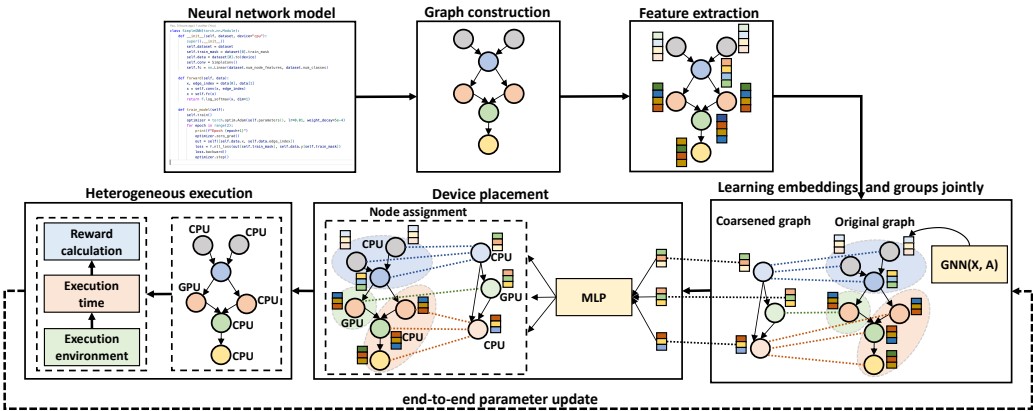

Figure 1: Overview of the proposed framework, HSDAG. **Graph construction.** We first convert a neural network model $c$ into a computation graph $G$, $repr : c \rightarrow G$. **Feature extraction.** Then, we calculate the initial feature matrix $\mathbf{X}^{(0)}$ capturing local and global connectivity information, node-aware features, information about the order of the nodes as well as features from fractal analysis. **Learning embeddings and groups jointly.** We further enrich node features $X^{(0)}$ using a GNN $: G \rightarrow \mathbf{Z}$ model and learn how to pool a graph $G$ jointly using a graph parsing network. In that way, we bridge the gap between grouper-placer and encoder-placer methods for device assignment. **Device placement.** A learnable MLP model classifies the nodes $V'$ of the coarsened graph $G' = (V', E')$ to the available devices $\mathcal{D}$. **Heterogeneous execution.** We map the device placement of $V'$ to $V$ based on the node assignment matrix $\mathcal{X}$ and apply the placement of all the operations into the execution environment to measure the execution time with the corresponding reward. **End-to-end parameter update.** We update our policy $\pi$ parameters $\theta$, i.e. the combination of GNN and MLP, based on the reward and renew the node feature matrix $\mathbf{Z}$ with the current cluster information. The entire framework supports end-to-end parameter updates and training.

where $r^*_{\pi, P} \in \mathbb{R}$ is the reward by following policy $\pi$ and placement $P$. We use a GNN (although any model can be employed) to learn the optimal policy denoted by $\pi^*_\theta$ and its set of parameters $\theta$. Let $l_P(G) \in \mathbb{R}^+$ denote the execution time of the computation graph $G$ by following the placement $P$. Our goal is to minimize execution time by learning the parameters

$$\theta^* = \arg\min_{\pi, \theta} l(G; \pi, \theta), \tag{2}$$

for the policy $\pi$ that yields the best results.

## 2.2 Graph construction

Given a set of neural network models $C = \{c_1, c_2, ..., c_{|C|}\}$, the first step is to decide a graph-based code representation $repr$ that converts a structure of a neural network model $c_i$ into a graph $G_i$, $repr : c_i \rightarrow G_i$. There is a list of different graph-based representations of neural network models including abstract syntax trees, contextual flow, control, data flow and LLVM IR graphs [17]. Although such graphs may contain valuable information and capture the latent information flow of a program, they tend to add unnecessary complexity to the overall process. Instead, for the experiments of this paper, we opt for representing the code of a neural network model $c_i$ as a computation graph, due to its expressiveness, simplicity, and practicality. A computation graph is generally smaller than its counterparts and could be easily allocated to specific devices. For instance, one can use several popular libraries to produce the computation graph $G_i$ of a given neural network model $c_i$. In this paper, we use the OpenVINO toolkit to generate the computation graphs as it generates smaller, already coarsened graphs compared to those of TensorFlow or PyTorch. Further information about the OpenVINO toolkit and examples of computation graphs are available in Appendix F. Even though it is optional, further coarsening can be performed using common co-locating operations or heuristics [22]. Such co-locating heuristics eliminate certain execution failures due to placement rule violations. In our experiments, we apply a simple algorithm for co-location to further condense the model into a smaller computation graph [22]. The graph construction step enables our approach to utilize graph representation learning techniques. Note that our framework is flexible and can

be coupled with any type of graph code representation for neural networks capable of producing a directed and acyclic graph $G$, similar to those of the computation graph representation.

## 2.3 Feature extraction

The proposed framework is also versatile as far as the initial node features are concerned. During our experimentation with various feature combinations (see Ablation studies in Section 3), we found that a mixture of features capturing local and global connectivity information, features from fractal analysis as well as node-specific features (e.g. topology, the order of a node and node type) leads to better results.

**Local structural features.** Specifically, the initial feature vector $\mathbf{x}_v^{(0)}$ of a node $v$ incorporates information about the node (operation) type $t_v$, in-degree $\delta_v^{in} = |\mathcal{N}_{in}(v)| = \sum_{u \in V} A(u, v) \in \mathbb{N}^+$ and out-degree $\delta_v^{out} = |\mathcal{N}_{out}(v)| = \sum_{u \in V} A(v, u) \in \mathbb{N}^+$ of $v$; here, $\mathcal{N}_{in}(v)$ and $\mathcal{N}_{out}(v)$ represent the sets of in-neighbors and out-neighbors of a node $v$, respectively. Initially, we use a one-hot encoding to embed each unique operation type $i$ into a tensor $T_i \in \{0, 1\}^{|T|}$, where $|T| \in \mathbb{N}^+$ is the number of unique operation types among all the input models $C$. Formally, for each operation type $t \in T = \{1, \ldots, |T|\}$, the one-hot encoding is defined as

$$T_i = \begin{cases} 1 & \text{if } i = t \\ 0 & \text{otherwise} \end{cases}, \; i \in \{1, \ldots, |T|\} \tag{3}$$

Similarly, we one-hot encode each unique in-degree $\delta_i^{in}$ and out-degree $\delta_j^{out}$ values into the $\Delta_i^{in} \in \{0, 1\}^{|\Delta^{in}|}$ and $\Delta_i^{out} \in \{0, 1\}^{|\Delta^{out}|}$ tensors, where $|\Delta^{in}| \in \mathbb{N}^+$ and $|\Delta^{out}| \in \mathbb{N}^+$ is the number of unique in-degree and out-degree values, respectively.

**Global structural features.** Relying solely on local features might miss capturing important global properties of the network. To capture the multi-scale structural properties of the network, we calculate the fractal dimension for each node $v \in V$. The fractal dimension $D(v)$ [28, 30] of a node $v$ is computed based on the mass distribution. Given the set of distances $\{r_1, r_2, \ldots, r_m\}$ from node $v$ to other nodes in the network, the fractal dimension $D(v)$ is calculated as follows:

$$D(v) = \frac{\sum_{k=1}^{m} (\log(r_k) - \bar{\log}(r))(\log(N(v, r_k)) - \log(\bar{N}(v, r)))}{\sum_{k=1}^{m} (\log(r_k) - \bar{\log}(r))^2} \tag{4}$$

where $r_k$ represents each distance in the set, $N(v, r_k)$ is the number of nodes within the distance $r_k$, and $\bar{\log}(r), \log(\bar{N}(v, r))$ are the mean values of $\log(r_k)$ and $\log(N(v, r_k))$, respectively.

**Positional features.** In an attempt to inject information about the order of the nodes, we associate each node $v$ with an integer $pos$ that encodes the topological order of the graph. To do so, we use a bijective mapping function $id : V \rightarrow \{1, \ldots, |V|\}$. Formally, if $v_i$ is the $i$-th node in the topological order, then $id(v_i) = i$. This kind of feature can be further enhanced using a function for positional encoding $PE : \mathbb{R} \times \mathbb{R} \rightarrow \mathbb{R}$:

$$PE(pos, k) = \begin{cases} \sin\left(\frac{pos}{10000^{\frac{2i}{d_{pos}}}}\right) & \text{if } k = 2i \\ \cos\left(\frac{pos}{10000^{\frac{2i}{d_{pos}}}}\right) & \text{if } k = 2i + 1 \end{cases}, \quad i \in [0, \frac{d_{pos}}{2}] \tag{5}$$

where $d_{pos}$ is the size of the embedding of feature $pos$.

**Node-specific features.** For each node $v$, we define the padded, fixed-size output shape tensor $S_v \in \mathbb{R}^{|S|}$, which is also provided as a piece of information in the original computation graph. Each digit of the output shape of a node $v$ is represented as a new dimension in the tensor $S_v$. We traverse the entire graph $G$ and obtain the maximum data output shape $|S| = \max_{v \in V}$ output_shape$(v)$.

Finally, we concatenate all the information for each individual node $v$ and form a node feature vector $\mathbf{x}_v^{(0)} \in \mathbb{R}^d$, where $d = T\|S\|\Delta^{in}\|\Delta^{out}\|D\|d_{pos}$. Building on top of the initial feature matrix $\mathbf{X}^{(0)} \in \mathbb{R}^{|V| \times d}$, in the next steps, we extend the representation learning capabilities of our framework

with recent techniques from the field of GNNs. While GNNs offer powerful feature extraction methods, this step is crucial as it provides a model with information about the nodes and structure of the graph, thereby accelerating the convergence of the process.

## 2.4 Learning embedding and groups jointly

We further enrich node features $\mathbf{X}^{(0)}$ and learn how to partition a given graph $G$ into an unspecified varying number of groups by employing the Graph Parsing Network (GPN) [24]. Existing grouper-placer methods typically operate with a predefined number of clusters during device placement exploration. They also employ non-trainable algorithms for graph partitioning or pooling relying mostly on human intuition and heuristics to group the nodes of a graph. This ad-hoc presetting of the group number leads to suboptimal solutions, which in turn inhibit the exploration and learning process of the overall framework. Instead, our framework treats both the number of node groups and the pooling algorithm as learnable parameters, which are trained in an end-to-end fashion. This step consists of three components: (1) graph and node encoding, (2) edge score matrix calculation and (3) graph partitioning and pooling.

**Graph and node encoding.** The graph and node encoding component is compatible with any neural network model and generates a node embedding $\mathbf{z}_v \in \mathbb{R}^{d'}$ for each node $v$, where $d'$ is the dimension of the node feature vector. In practice, we use an embedding function $\text{GNN} : G \rightarrow \mathbf{Z} \in \mathbb{R}^{|V| \times d'}$ as our main graph encoder; self-supervised techniques may also be employed to pre-train the embedding function GNN, which aids in the downstream task of device placement [32]. As a result of using a GNN as an encoder, the learnable feature matrix $\mathbf{Z} \in \mathbb{R}^{|V| \times d'}$ captures both node- and structure-aware information about the graph $G$. As we mentioned before, our framework is model-agnostic and allows for utilizing different GNN functions. In the interest of clarity, we formulate the representation learning step using a GCN [14] model with a single graph convolutional layer:

$$\mathbf{Z} = \text{GNN}(\mathbf{X}, A) = \sigma(\hat{\mathbf{D}}^{-1/2} \hat{\mathbf{A}} \hat{\mathbf{D}}^{-1/2} \mathbf{X}^{(0)} \mathbf{W}) \in \mathbb{R}^{|V| \times d'} \tag{6}$$

where $\hat{\mathbf{A}} = \mathbf{A} + \mathbf{I} \in \{0, 1\}^{|V| \times |V|}$ denotes the adjacency matrix with self-loops and $\hat{\mathbf{D}}_{ii} = \sum_{j=0} \hat{\mathbf{A}}_{ij}$ is the corresponding diagonal degree matrix, $\mathbf{W} \in \mathbb{R}^{d \times d'}$ is a matrix of learnable parameters, $\mathbf{X}^{(0)} \in \mathbb{R}^{|V| \times d}$ is a matrix with the input features of each node $v$ and $\sigma(\cdot)$ denotes an activation function such as $\text{ReLU}(\cdot) = \max(0, \cdot)$.

**Edge score matrix calculation.** This component accepts any differentiable neural network model to calculate an edge score matrix $\mathcal{S} \in [0, 1]^{|V| \times |V|}$. Given an edge $e$ connecting two nodes $v, u$ and their embeddings $\mathbf{z}_v, \mathbf{z}_u$, then the score $\mathcal{S}_{v,u} = \mathcal{S}_e$ is calculated as follows:

$$\mathcal{S}_{v,u} = \sigma(\phi(\mathbf{z}_v \odot \mathbf{z}_u)) \in [0, 1] \quad s.t. \quad \mathcal{S} = \mathcal{S} \odot A \tag{7}$$

where $\sigma(x) = \text{sigmoid}(x) = \frac{1}{1 + e^{(-x)}}$ and $\phi$ can be any differentiable neural network. During our experimentation, we found that setting $\phi = \text{MLP}$ yields good performance w.r.t. the task of device placement. The magnitude of an edge score $\mathcal{S}_e$ quantifies the strength of the relationship between the connected nodes $v$ and $u$. A higher edge score $\mathcal{S}_e$ implies a stronger relationship, increasing the probability for the nodes $v$ and $u$ to be grouped into the same partition $\mathcal{P}_i \in \mathcal{P}$. Formally, this can be expressed as:

$$P(v \in \mathcal{P}_i \wedge u \in \mathcal{P}_i) \propto \mathcal{S}_e, \quad e = (v, u) \tag{8}$$

As a result, the higher the edge score $\mathcal{S}_e$, the higher the probability that nodes $v$ and $u$ will be grouped into the same partition, reflecting their relational affinity.

**Graph partitioning and pooling.** The graph partitioning and pooling component uses the computed edge scores $\mathcal{S}$ to partition the entire graph $G$. Specifically, it iterates through each node $v$ in the graph $G$ and identifies the edge with the highest score among all edges connected to that node. In a graph with $|V|$ nodes, this process may identify up to $|\mathcal{E}| \leq |V|$ such edges. Only these $|\mathcal{E}|$ edges are retained and the remaining edges are discarded, automatically dividing the graph into multiple groups. The set of the remaining edges is then defined as:

$$\mathcal{E} = \{(v, u) | v \in V, u = \arg \max_{u' \in \mathcal{N}_{(v)}} \mathcal{S}_{v,u'}\} \tag{9}$$

This grouping method ensures that nodes within each group have stronger local connectivity and tighter relationships, making them more suitable for being assigned to the same device for execution.

The final step is to create a node assignment matrix $\mathcal{X} \in \mathbb{R}^{|V| \times |V'|}$ that maps each node $v$ in the original graph $G$ to a node $v'$ in the coarsened graph $G' = (V', E')$. To construct the node assignment matrix $\mathcal{X}$ we use the graph parsing algorithm $\mathcal{A}$, as proposed in [24]:

$$\mathcal{X} = \mathcal{A}(\mathcal{E}) \tag{10}$$

The adjacency matrix $A' \in \{0,1\}^{|V'| \times |V'|}$ of the pooled graph $G'$ is then defined as:

$$A' = \mathcal{X}^T \cdot A \cdot \mathcal{X} \tag{11}$$

### 2.5 Reinforcement learning for node-based device assignment

In this step, we combine the GPN from the previous component and an MLP to learn a policy $\pi : G' \to P'$. After we obtain the device placement $P'$, we use the node assignment matrix $\mathcal{X}$ to map each node $v'$ of the coarsened graph $G'$ to a node $v$ of the original graph $G$. In that way, we manage to assign a device $p_v$ for each node $v$ of the graph $G$. At each RL episode, we infer the machine learning model with the updated operation device placement $P'$ and get the inference latency $l_{P'}(G')$. Our ultimate goal is to choose a reward function that maximizes the reward when the latency is low. Thus, we use the reward function $r_{P'}(G') = \frac{1}{l_{P'}(G')}$. To find the proper stochastic group detection and placement policy parameters $\theta$, we maximize the objective function

$$J(\theta) = \mathbb{E}_{P \sim \pi(P|G';\theta)}[r(P, G')] \tag{12}$$

For each time step, we update the node embedding $\mathbf{Z}_v$ of a node $v$ by summing up the embedding $\mathbf{Z}_{v'}$ of its corresponding coarsened node $v'$: $\mathbf{Z}_v = \mathbf{Z}_v + \mathbf{Z}_{v'}$. We then form a new graph $G'$ which can also be considered as the new state. We then run a new round of representation and group learning (Section 2.4) and device placement for the new graph $G'$. We update our policy parameter gradient by REINFORCE [27] using the Adam [13] optimizer producing:

$$\nabla_\theta J(\theta) = \mathbb{E}_{P \sim \pi(P|G';\theta)}[r(P, G) \cdot \nabla_\theta \log p(P \mid G'; \theta)] \tag{13}$$

We record $x$ steps in the buffer and compute the reward of each device placement. After $x$ steps, we update the policy parameter with the cumulative reward and loss

$$\nabla_\theta J(\theta) \approx - \sum_{i=1}^{x} \nabla_\theta \log p(P \mid G'; \theta) \cdot \gamma^i \cdot r(P_i, G) \tag{14}$$

where $\gamma$ is the discount rate for the reward at the current time step to the previous time steps.

## 3 Experiments

### 3.1 Benchmarks

To evaluate our approach we use the computation graphs created from three popular benchmarks: (1) **Inception-V3**: The Inception-V3 architecture [25] is extensively employed for image recognition and visual feature extraction [12]. This neural network consists of multiple blocks, each comprising various branches of convolutional and pooling layers. These branches are capable of parallel execution and are concatenated to form inputs for the subsequent block. However, the depth of the network limits this parallelism since later blocks must wait for the completion of earlier ones; (2) **ResNet**: ResNet [10] is a widely-used model for image classification. It consists of multiple convolutional layers and uses residual connections to reduce the effects of the vanishing gradient problem. We use the ResNet-50, which is a 50-layer convolutional neural network; (3) **BERT**: BERT [6] is a language model relying on the transformer architecture. It pre-trains deep bidirectional representations on unlabeled data jointly. It can be used as the base to fine-tune models for a list of tasks, including question answering and language inference. Several versions of the BERT model exist. In this paper, we use the base uncased version. Important statistics and a more detailed description of the benchmark models are available in Table 1 and Appendix D.

### 3.2 Setup

We implement the variant of HSDAG and baseline models with the PyTorch Geometric framework [7]. The Adam [13] optimization algorithm is used for the optimization of the parameters of the models. We run our experiments on real hardware using the OpenVINO toolkit version 2023.3.0 [2].

---

[2] https://github.com/openvinotoolkit

Table 1: Statistics of computation graphs of the benchmarks used in our experiments. $|V|$: the number of nodes, $|E|$: the number of edges, $\bar{d}$: the average degree.

| BENCHMARK | $|V|$ | $|E|$ | $\bar{d}$ |
|---|---|---|---|
| INCEPTION-V3 | 728 | 764 | 1.05 |
| RESNET | 396 | 411 | 1.04 |
| BERT | 1009 | 1071 | 1.06 |

**Devices.** The available devices for our experiments are the following: **(1)** CPU: 12th Gen Intel(R) Core(TM) i9-12900K, **(2)** GPU.0: Intel(R) UHD Graphics 770 (iGPU) and **(3)** GPU.1: Intel(R) Data Center GPU Flex 170 (dGPU). Our server has 64GB of memory.

### 3.3 Baseline comparison

Aiming to test the performance of the proposed framework, we evaluate the proposed variant against a list of state-of-the-art baseline methods. The selected baseline models may differ from the variant of our framework in many ways, in terms of their architecture and the algorithms they employ to implement their components. Furthermore, they may ignore parts of the proposed framework or implement them differently (e.g., learn node embeddings and clusters separately). The main purpose of the baselines is the evaluation of our framework w.r.t. the task of device placement.

1. **CPU-only**. It assigns the entire computation graph to CPU. It does not include any part of the proposed approach, except for the device assignment component.

2. **GPU-only**. It assigns the entire computation graph to GPU. Similar to CPU-only, the device assignment part of our framework is the only one that is implemented.

3. **OpenVINO-CPU**. This baseline method lets the OpenVINO optimization toolkit decide whether the entire computation graph should be assigned to CPU or GPU, with CPU set as the first preference.

4. **OpenVINO-GPU**. A baseline similar to OpenVINO-CPU with GPU set as the first preference.

5. **Placeto** [1]. It uses GNNs to learn features for any computation graph as proposed in [1]. Therefore it enables the transfer of a learned device placement policy to new computation graphs without further re-training.

6. **RNN-based approach** [22]. An RL framework trained to optimize device placement by utilizing a sequence-to-sequence LSTM model and a content-based attention mechanism.

Table 2 shows the performance of the compared models as far as the task of device placement and the reduction in execution (inference) time are concerned. On Inception-V3, our framework achieves a $17.9\%$ speedup over the CPU-only baseline, reducing the inference time from $0.0128$ seconds to $0.0105$ seconds. This performance surpasses other baselines, such as GPU-only ($6.25\%$ speedup) and Placeto ($9.38\%$ speedup). Similarly, on ResNet, our framework delivers a $52.1\%$ speedup, reducing the inference time to $0.00766$ seconds, which is significantly better than the GPU-only ($51.2\%$ speedup) and OpenVINO-GPU ($45.3\%$ speedup) baselines. The most substantial improvement is observed on the BERT benchmark, where our framework achieves a $58.2\%$ speedup, reducing the inference time to $0.00267$ seconds, outperforming the GPU-only baseline ($56.5\% speedup$). These results highlight the efficiency and effectiveness of the proposed device placement approach, leveraging graph coarsening, node representation learning, and reinforcement learning to optimize computation graph execution on heterogeneous hardware environments.

### 3.4 Ablation studies

To understand the impact of the components, steps and configurations of HSDAG, we conduct an ablation study. Various modifications to the framework were tested, such as removing graph structural features, output shape features, and node IDs. Overall, the results are shown in Table 3. indicate that each of these components plays a significant role in achieving optimal performance.

Table 2: Evaluation results of different models on the device placement task. The **best results** for each baseline model across benchmarks are highlighted in **bold**. $l_P(G)$ denotes the execution time (in seconds) for each model. Speedup % denotes the speedup with respect to the CPU-only baseline. On the execution time $l_P(G)$ column, lower ($\downarrow$) scores are better. On the Speedup % column, higher ($\uparrow$) scores are better. To get accurate results, we measure the inference time with the same device displacement 10 times and take the average of the last 5 measurements. OOM: out of memory.

| | Inception-V3 | | ResNet | | BERT | |
| --- | --- | --- | --- | --- | --- | --- |
| | $l_P(G)$ | Speedup % | $l_P(G)$ | Speedup % | $l_P(G)$ | Speedup % |
| CPU-only | 0.0128 | 0 | 0.0160 | 0 | 0.00638 | 0 |
| GPU-only | 0.0120 | 6.25 | 0.00781 | 51.2 | 0.00277 | 56.5 |
| OpenVINO-CPU | 0.0128 | 0 | 0.0234 | $-46.3$ | 0.00657 | $-2.98$ |
| OpenVINO-GPU | 0.0138 | $-7.81$ | 0.00876 | 45.3 | 0.00284 | 55.5 |
| Placeto | 0.0116 | 9.38 | 0.00932 | 41.8 | 0.00651 | $-2.04$ |
| RNN-based | 0.0128 | 0 | 0.00875 | 45.3 | OOM | OOM |
| HSDAG | **0.0105** | **17.9** | **0.00766** | **52.1** | **0.00267** | **58.2** |

Table 3: Results of the framework variants of the ablation study on the device placement task. $l_p(G)$ denotes the execution time (in seconds) for each model. Speedup % denotes the speedup with respect to the CPU-only baseline. On the execution time $l_p(G)$ column, lower ($\downarrow$) scores are better. On the Speedup % column, higher ($\uparrow$) scores are better. To get accurate results, we measure the inference time with the same device displacement 10 times and take the average of the last 5 measurements.

| | Inception-V3 | | ResNet | | BERT | |
| --- | --- | --- | --- | --- | --- | --- |
| | $l_P(G)$ | Speedup % | $l_P(G)$ | Speedup % | $l_P(G)$ | Speedup % |
| CPU-only | 0.0128 | 0 | 0.0160 | 0 | 0.00638 | 0 |
| Original | 0.0105 | 17.9 | 0.00766 | 52.1 | 0.00267 | 58.2 |
| w/o output shape | 0.0117 | 8.59 | 0.00768 | 52.0 | 0.00278 | 56.4 |
| w/o node ID | 0.0117 | 8.59 | 0.00768 | 52.0 | 0.00279 | 56.4 |
| w/o graph structural features | 0.0109 | 14.8 | 0.00766 | 52.1 | 0.00268 | 58.2 |

**No graph structural features.** For each node of $v$ in the computation graph $G$, we ignore features from fractal analysis, in-degree, and out-degree. Removing graph structural features clearly impacts the framework's ability to capture the global and local structural information of the computation graph. The results from this ablation show a decrease in performance, although not as important as other feature removals. For example, the inference time speedup on Inception-V3 drops to $14.8\%$ from $17.9\%$ when these features are excluded. This indicates the importance of these features in capturing the hierarchical and interconnected nature of computation graphs.

**No output shape features.** We assume that the output data shape of each operation reflects the computation requirement for the corresponding operation. We do not include the output data shape in the node feature vector to test its effectiveness. We observe a significant decrease in performance. The speedup for Inception-V3 drops from $17.9\%$ to $8.59\%$. This suggests that the features related to the output shape are crucial for understanding the computational load associated with each node in the graph. Without this information, the framework's ability to optimize device placement is compromised, leading to less efficient computation graph execution.

**No node ID.** In this case, we do not use the node ID to encode the topological sequence of the node $v$ in a given computation graph $G$. Omitting information about the node ID results in a significant performance drop. The speedup for Inception-V3 is reduced to $8.59\%$. This highlights the critical role of node IDs in preserving the order and dependencies within the computation graph. The framework lacks essential information about the execution order of operations, leading to sub-optimal device placement and reduced overall efficiency.

### 3.5 Downstream Model Performance and Runtime Complexity

As a sanity check for our method, we show the performance on downstream tasks is not affected. Theoretically, since the end-to-end training pipeline itself does not change, we do not expect any

impact on the performance. Nonetheless, we show experimental results on 3 exemplar cases to empirically verify statement.

**Inception-V3**: We performed image classification inference on images depicting Samoyed dogs. All the parameters are directly derived from the torchvision pre-trained model. We did not change any configuration on the data type of the model. The classification accuracy of Inception-V3 using the best device placement is **82.77**%. For the GPU-only experiments the classification accuracy is **82.72**%. For the CPU-only experiments the classification accuracy is **82.33**%.

**ResNet**: Similarly, we performed image classification inference using the ResNet model on the same dataset. The classification accuracy with the best device placement is **45.37**%. For the GPU-only experiments the classification accuracy is **45.37**%. For the CPU-only experiments the classification accuracy is **45.44**%.

**BERT**: We evaluated the performance of the BERT model using the output embeddings from the different device placements. We calculated their mean squared error, cosine similarity and Euclidean distance (MSE: the lower the better, Cosine Similarity: the higher the better, euclidean distance: the lower the better). The results are shown on Table 4.

Table 4: BERT performance on downstream tasks.  Table 5: Empirical runtime complexity comparison.

| Comparison | MSE | CS | L2 norm | | Model | Inception-V3 | ResNet | BERT |
|---|---|---|---|---|---|---|---|---|
| CPU vs GPU | 3.049e-05 | 0.999 | 0.432 | | Placeto | 2808s | 1162s | 4512s |
| **CPU vs HSDAG** | **6.819e-07** | **0.999** | **0.064** | | RNN-based | 3706s | 1212s | OOM |
| GPU vs HSDAG | 3.174e-05 | 0.999 | 0.441 | | HSDAG | **2454s** | **1047s** | **2765s** |

These experiments empirically demonstrate that HSDAG does not affect the performance of the model in the downstream tasks. All models have similar performance regardless of the running device (e.g. CPU, GPU or heterogeneous device). Finally, we also conducted an empirical runtime complexity estimation in order to show HSDAG's superiority in terms of execution time. We compare against the Placeto and the RNN-based device placement methods for the same 3 models. The results are shown in Table 5.

## 4  Conclusion

We introduce a flexible framework for the task of device placement. Our framework relies on smaller computation graphs and is divided into five steps such as graph coarsening, node representation learning and policy optimization using reinforcement learning. It supports end-to-end model training and is aware of the directed and acyclic nature of the computation graphs under consideration. Additionally, we propose a model variant that facilitates graph representation learning and personalized graph partitioning jointly with an unspecified number of node clusters. The experimental results highlighted that our framework is flexible, and robust and mitigates the shortcomings of the grouper-placer and encoder-placer models by capitalizing on the best aspects of the two worlds. The suggested placements improve the inference speed for the benchmark models compared to widely used baselines. One interesting future work direction is to explore different RL problem formulations as well as different reward structures, such as the incremental rewards used in [1]. Another direction is to study the interplay of the reward and generalizability as well as potentially using reward models rather than measuring reward. This could unlock a more efficient algorithm as the current setup relies on measuring the inference latency, which has practical limitations.

**Limitations.** The latency measurements did not consider the temperature change of the environment of the system and surroundings. During our experiments, we allocate the CPU for both experiment and policy due to the limitation of the setup. At the same time, iGPU is not considered throughout the experiment because we consider it to be always slower than both CPU and GPU. We attempted to obtain the source code for the baseline methods, but they were not made available from the corresponding authors. As such, we implemented and reproduced the baseline methods with our best effort from their published papers.

## Acknowledgements

The University of Southern California acknowledges the support by the U.S. Army Research Office (ARO) under Grant No. W911NF-23-1-0111, the National Science Foundation (NSF) under the Career Award CPS-1453860, CCF-1837131, MCB-1936775, CNS-1932620 and the NSF award No. 2243104 under the Center for Complex Particle Systems (COMPASS), the Defense Advanced Research Projects Agency (DARPA) Young Faculty Award and DARPA Director Award under Grant Number N66001-17-1-4044, National Institutes of Health (NIH) under grant R01 AG 079957, an Intel faculty award and a Northrop Grumman grant. The views, opinions, and/or findings in this article are those of the authors and should not be interpreted as official views or policies of the Department of Defense, the National Institutes of Health, or the National Science Foundation.

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

## A    Code availability

The source code is available at .

## B    Related work

A plethora of methods have been developed to optimize device placement [21, 33, 1]. These methods can be categorized as (1) heuristic methods or (2) methods relying on RL techniques and deep learning. Heuristic methods are able to identify relatively optimal device placement solutions in a short time, but they require ad-hoc adjustments on different devices. Furthermore, they often fail to achieve the best outcomes [21]. In contrast, using RL for device placement generates better placement schemes for various devices.

The work in [22] first proposed an RL framework trained with the REINFORCE algorithm to optimize device placement, utilizing an attentional sequence-to-sequence model with LSTM and a content-based attention mechanism. Building on [22], the authors in [21] introduced a feed-forward neural network as a 'grouper', transforming the simple sequence-to-sequence model into a hierarchical model. This hierarchical model enhances the grouping ability of the RL framework, facilitating the rapid identification of more suitable placement solutions.

As opposed to using sequence-to-sequence models as the agent of the RL framework, Placeto [1] employed graph neural networks to extract and learn essential features of computation graphs, enabling the transfer of a learned placement policy to unseen computation graphs without the need for retraining. The proposed approach used the structural information of the graph to better understand the interdependencies and communication costs, leading to more informed placement decisions. However, its node-by-node placement methodology might lead to longer training times and difficulties in scaling to very large computational graphs. The work in [33] combined graph neural networks and sequence-to-sequence models into an RL framework, initially encoding the computation graphs using GraphSAGE [8], then placing operations onto devices in one shot with the Transformer-XL model [5]. Leveraging the combination of GNNs and sequence-to-sequence models, the authors in [15] adopted contrastive learning as a pretraining method for GNNs, potentially impacting the training process of the RL framework. They introduced a GNN model that incorporates both the graph structure of the computation task and the characteristics of the hardware devices. Their model predicts the execution time of various operations across different devices, providing a holistic view that guides the placement algorithm.

**Relevance to our approach**. According to [33], all device placement frameworks relying on RL can be split into two main categories: 'grouper-placer' and 'encoder-placer'. The former structure initially divides the entire computation graphs into multiple groups and then allocates devices for each group. The latter learns features of computation graphs in the first phase and then predicts the device for each operation within the computation graphs based on the extracted features. The authors in [33] stated that the 'encoder-placer' structure offers more flexibility and generality than traditional 'grouper-placer' designs. The two structures share the same concept of grouping computation graphs with their main difference being that the 'grouper-placer' structure explicitly defines group partitioning, whereas the 'encoder-placer' structure achieves the concept of grouping through feature grouping encoding. Therefore, previous works essentially employed the technique of grouping and achieved encouraging results. Following a similar direction, our framework builds on the concept of grouping and utilizes a GNN as an agent to learn a policy for the device placement problem.

## C    Preliminaries

**Graph**. Consider a graph $G = (V, E)$ where $V$ denotes the set of vertices and $E \subseteq V \times V$ represents the edges connecting these vertices. Each vertex $v \in V$ is associated with a feature vector $\mathbf{x}_v$. Edges can also have feature vectors denoted by $\mathbf{x}_{uv} \forall (u, v) \in E$, representing the features of the edge from node $u$ to node $v$.

**Neighborhood of a node**. The neighborhood of a node $v$ is defined as follows:

$$\mathcal{N}(v) = \mathcal{N}_{in}(v) \cup \mathcal{N}_{out}(v) = \{u \in V : (u, v) \in E \vee (v, u) \in E\} \tag{15}$$

## C.1 Message-Passing neural networks

The core operation of GNNs involves a message-passing mechanism where nodes communicate and update their states. The iterative update process involves several layers and can be formulated as follows.

**Message function**. For each edge $(u, v)$ a message $\mathbf{m}_{uv}^{(t)}$ is computed at each layer $t$. This message is a function of the features of the adjacent nodes and the edge itself:

$$\mathbf{m}_{uv}^{(t)} = \text{MESSAGE}(\mathbf{h}_u^{(t)}, \mathbf{h}_u^{(t)}, \mathbf{x}_{uv}) \tag{16}$$

where $\mathbf{h}_u^{(t)}$ and $\mathbf{h}_v^{(t)}$ are the feature vectors of nodes $u$ and $v$ at layer $t$.

**Aggregation function**. Each node $v$ aggregates messages from its neighborhood of nodes $\mathcal{N}(v)$:

$$\mathbf{a}_v^{(t+1)} = \text{AGGREGATE}(\{\mathbf{m}_{uv}^{(t)} : u \in \mathcal{N}(v)\}) \tag{17}$$

**Update function**. The feature vector of each node $v$ is updated based on its previous state and the aggregated messages:

$$\mathbf{h}_v^{(t+1)} = \text{UPDATE}(\mathbf{h}_v^{(t)}, \mathbf{a}_v^{(t+1)}) \tag{18}$$

In all the cases above, UPDATE, AGGREGATE and MESSAGE can be any arbitrary differentiable function (i.e., neural networks).

**Layer and model configuration.** A GNN consists of several layers, where each layer applies the message, aggregation and update functions iteratively. The output $\mathbf{h}_v^{(T)}$ after $T$ layers can be utilized for various tasks such as classification, regression or other predictive tasks relevant to the nodes or the entire graph. In our case, we use the output $\mathbf{h}_v^{(T)}$ for the device placement problem.

# D Benchmark models

## D.1 Inception-V3

Inception-V3 [25] is an advanced convolutional neural network (CNN) that has significantly contributed to the field of image recognition. Building on the architecture of its predecessors, Inception-V3 incorporates several key innovations to enhance both accuracy and computational efficiency. The model utilizes factorized convolutions, which decompose traditional convolutions into smaller, more efficient operations, effectively reducing the computational burden without compromising performance. Additionally, Inception-V3 employs auxiliary classifiers at intermediate layers to combat the vanishing gradient problem, thereby improving the training process. The model also extensively uses batch normalization to stabilize and accelerate convergence. These enhancements enable Inception-V3 to achieve state-of-the-art results on large-scale image datasets such as ImageNet, making it a robust and efficient choice for a variety of computer vision tasks, including image classification and object detection.

## D.2 ResNet

ResNet (Residual Networks) [10] represents a groundbreaking advancement in the field of deep learning, particularly in the domain of image recognition. ResNet addresses the challenge of training very deep neural networks by incorporating residual learning. The key innovation lies in its use of residual blocks, which allow the network to learn residual functions with reference to the input layer. This approach helps mitigate the vanishing gradient problem, enabling the training of networks with hundreds or even thousands of layers without degrading performance. Each residual block contains skip connections or shortcuts, that bypass one or more layers, facilitating better gradient flow and making the optimization of deep networks more feasible. ResNet models have achieved remarkable success in various competitions, including the ImageNet Large Scale Visual Recognition Challenge, where they set new benchmarks for accuracy. The architecture's robustness and scalability have made ResNet a preferred choice for numerous computer vision tasks, from image classification and object detection to semantic segmentation and beyond.

## D.3 BERT

BERT (Bidirectional Encoder Representations from Transformers) [6] is a transformative model in NLP. Unlike traditional NLP models that process text either from left to right or right to left, BERT utilizes a bidirectional approach, allowing it to consider the full context of a word by looking at the words that come before and after it. This bidirectional capability is achieved through the use of Transformer encoders, which leverage self-attention mechanisms to weigh the importance of different words in a sentence relative to each other. One of BERT's key innovations is its pre-training and fine-tuning approach. BERT is first pre-trained on a large corpus of text using two unsupervised tasks: Masked Language Modeling (MLM) and Next Sentence Prediction (NSP). In MLM, some of the words in the input are masked at random, and the model is trained to predict these masked words based on their context. In NSP, the model is trained to predict if two sentences are consecutive. After pre-training, BERT can be fine-tuned on a variety of specific tasks such as question answering, sentiment analysis, and named entity recognition, by adding just a few additional output layers.

# E Algorithm Pseudocode

In this section, we present the detailed pseudocode for our main method, HSDAG.

---

**Algorithm 1** Hierarchical Structure-Aware Device Assignment Graph (HSDAG)

---

1: Initialize computation graph $\mathcal{G}$, coarsened graph $\mathcal{G}'$, node features $\mathbf{F}$, and maximum iterations $max_{interations}$.
2: Initialize parameters $\theta$ of Graph Parsing Network (GPN) and Multi-Layer Perceptron (MLP).
3: Initialize node assignment matrix $\mathbf{X}$, Adam optimizer, buffer size $x$, and discount factor $\gamma$.
4: **for** $t = 1$ to $max_{interations}$ **do**
5:     Apply GPN: $(\mathcal{C}, \mathbf{F}_c) = \text{GPN}(\mathcal{G}, \mathbf{F})$, where $\mathcal{C}$ are clusters and $\mathbf{F}_c$ are cluster features.
6:     Apply MLP: $\mathbf{P}' = \text{MLP}(\mathbf{F}_c)$ to get device placement for the coarsened graph.
7:     Map $\mathbf{P}'$ to original graph $\mathcal{G}$ using assignment matrix $\mathbf{X}$.
8:     Deploy $\mathcal{G}$ with device placement $\mathbf{P}'$ and measure inference latency $l_{p'}(G')$.
9:     Calculate reward: $r_{p'}(G') = 1/l_{p'}(G')$.
10:     Update embeddings: $\mathbf{Z}_v = \mathbf{Z}_v + \mathbf{Z}_{v'}$ for each node $v$ and its corresponding coarsened $v'$.
11:     Form a new coarsened graph $\mathcal{G}'$ as the new state.
12:     Run a new round of representation and group learning for a new graph $\mathcal{G}'$.
13:     Store step information $(\mathcal{P}, \mathcal{G}', r_{p'}(G'))$ in buffer.
14:     **if** buffer reaches $x$ steps **then**
15:         Compute gradient: $\nabla_\theta J(\theta) \approx -\sum_{i=1}^{x} \nabla_\theta \log p(P|G'; \theta) \cdot \gamma^i \cdot r(P_i, G)$
16:         Update policy parameters $\theta$ using Adam optimizer with the computed gradient.
17:         Clear buffer.
18:     **end if**
19:     **if** convergence criteria are met **then**
20:         Break
21:     **end if**
22: **end for**
23: **return** optimal device placement $\mathbf{P}_{opt}$.

---

The Graph Parsing Network algorithm is explained in detail in [24] and is reproduced in Algorithm 2 for convenience.

# F Computation graph construction using the OpenVINO toolkit

The OpenVINO (Open Visual Inference and Neural Network Optimization) toolkit, developed by Intel, is a comprehensive toolkit designed to accelerate the development of applications involving deep learning inference and computer vision. It enables developers to deploy pre-trained models across a variety of Intel hardware, including CPUs, integrated graphics, VPUs (Vision Processing Units), and FPGAs (Field Programmable Gate Arrays), thereby optimizing performance and efficiency.

**Algorithm 2** Graph Parsing Network

1: Initialize input graph $\mathcal{G}$ with node features $\mathbf{X}^{(0)}$ and adjacency matrix $\mathbf{A}$.
2: Initialize learnable parameters $\mathbf{W}$ for the GNN.
3: **for** each iteration **do**
4:     Perform graph and node encoding: $Z = GNN(X, A) = \sigma\left(\hat{D}^{-1/2}\hat{A}\hat{D}^{-1/2}X^{(0)}W\right)$ where $\hat{A} = A + I, \hat{D}_{ii} = \sum_{j=0}\hat{A}_{ij}$
5:     Calculate edge score matrix:$S_{v,u} = \sigma\left(\phi(z_v, z_u)\right)$ where $\phi$ is MLP.
6:     Apply Graph Parsing Algorithm $\mathcal{A}$:
7:         Stage 1: $\hat{C}^{(k)} \leftarrow DOM(C^{(k)})$
8:         Initialize: $s \leftarrow \emptyset, p \leftarrow 0$
9:         Stage 2: While $sum(\hat{C}^{(k)}) \neq 0$:
10:           $idx \leftarrow argmax(\hat{C}^{(k)})$
11:           $q \leftarrow |idx|$
12:           $(l, idx') \leftarrow EXP(idx, \hat{C}^{(k)})$
13:           $idx \leftarrow union(idx, idx')$
14:           While $|idx| = q$:
15:             $s \leftarrow union(s, (i, p)|i \in l)$
16:             $\hat{C}^{(k)}_{i,j} \leftarrow 0, \forall(i,j) \in idx$
17:             $p \leftarrow p + 1$
18:         Stage 3: Finalize clusters.
19:     Create node assignment matrix: $\mathcal{X} = \mathcal{A}(\mathcal{E})$
20:     Define adjacency matrix for pooled graph: $\mathbf{A}' = \mathcal{X}^T \cdot \mathbf{A} \cdot \mathcal{X}$
21: **end for**
22: **return** pooled graph $\mathcal{G}'$ and node assignment matrix $\mathcal{X}$

### F.1 Key Features and Components

- **Model Optimizer**: This component is a cross-platform command-line tool that facilitates the conversion of trained models from popular deep learning frameworks such as TensorFlow, Caffe, MXNet, Kaldi, and ONNX into an optimized Intermediate Representation (IR) format. This format consists of a pair of files (.xml and .bin) which describe the network structure and contain the binary weights, respectively. The Model Optimizer reduces the model size and increases inference speed by performing transformations such as batch normalization folding, constant folding, and layer fusion.

- **Inference Engine**: The Inference Engine is a high-performance, inference delivery library that provides an API to deploy the optimized models on various Intel hardware. It abstracts the complexity of hardware-specific optimizations and provides a unified API for different devices, allowing developers to write their code once and run it across different Intel hardware without modification. The Inference Engine supports heterogeneous execution, enabling the distribution of computational workloads across multiple devices.

- **Pre-Trained Models**: OpenVINO includes a Model Zoo that provides a collection of optimized and pre-trained models for a wide range of applications, including image classification, object detection, semantic segmentation, and more. These models can be used directly or fine-tuned for specific use cases, significantly reducing the time required to develop and deploy AI solutions.

- **Deep Learning Workbench**: This is a graphical interface that helps developers visualize, fine-tune, and optimize models. The Deep Learning Workbench provides tools for accuracy validation, performance benchmarking, and deployment configuration, streamlining the model optimization process.

- **Support for Heterogeneous Execution**: OpenVINO supports the deployment of deep learning models across heterogeneous hardware environments, enabling simultaneous use of CPUs, GPUs, VPUs, and FPGAs. This flexibility allows developers to maximize resource utilization and achieve optimal performance for their applications.

- **Extensive Documentation and Community Support**: Intel provides extensive documentation, tutorials, and sample applications to assist developers in leveraging the full capabilities

of OpenVINO. Additionally, an active community and support forum offer assistance and facilitate knowledge sharing among developers.

## F.2 Applications and Use Cases

OpenVINO is widely used in various industries, including healthcare, retail, automotive, and industrial automation. It enables applications such as medical imaging diagnostics, retail analytics, autonomous driving, and smart manufacturing. The toolkit's ability to optimize and deploy deep learning models on edge devices makes it particularly suitable for real-time, low-latency applications where efficient resource utilization is critical.

## F.3 Computation graphs of the benchmark models

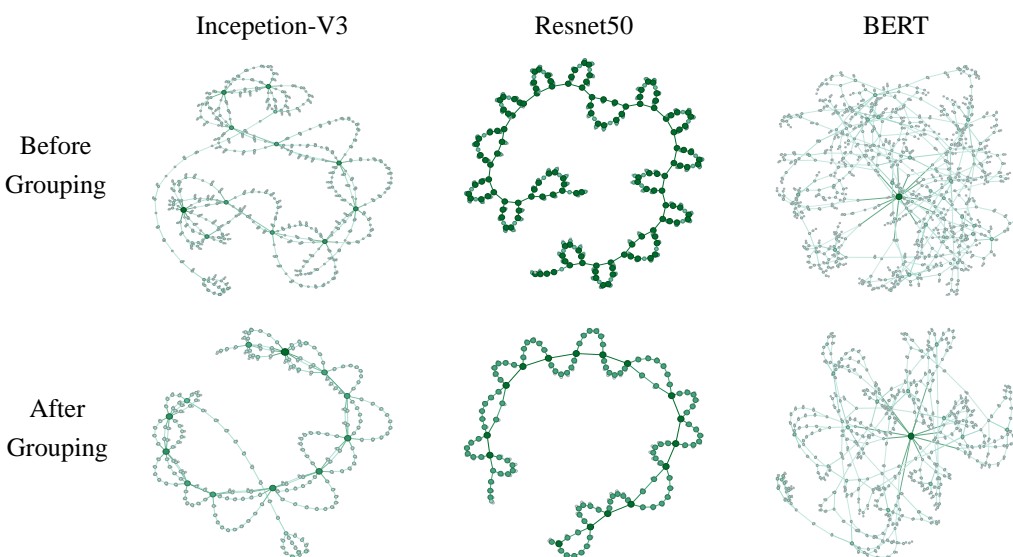

Figure 2: The computation graph of each of the benchmark models before and after the graph partitioning and pooling.

## G Co-location heuristic

Following the methodology outlined in [22], we implemented a co-location heuristic to coarsen the graph. For each vertex $v_i \in V$ considered in topological order, we applied the following criteria: if $v_j$ is the sole child neighbor of $v_i$, and simultaneously, $v_i$ is the sole parent neighbor of $v_j$, then $v_i$ and $v_j$ are grouped into the same co-location set $C_s$. These co-location sets were used to form a coarsened graph $CG$, with the operation type of each co-location set $C_s$ determined by the mean of the operation types of all $v \in C_s$.

## H Parameters and Hyper-parameters

This is section we provide some implementation details, namely several parameters and hyper-parameter choices, to aid in the reproducibility of our results. The meaning of each parameter is listed bellow as well.

**Parameter Descriptions**

- **num_devices**: Represents the number of processors used to run the model.
- **hidden_channel**: Represents the dimension of the node embeddings generated by GNN encoding.

Table 6: Model Parameters

| Parameter | Value | Parameter | Value |
|---|---|---|---|
| num_devices | 2 | dropout_parsing | 0.0 |
| hidden_channel | 128 | link_ignore_self_loop | True |
| layer_trans | 2 | activation_final | True |
| layer_gnn | 2 | learning_rate | 0.0001 |
| layer_parsingnet | 2 | max_episodes | 100 |
| gnn_model | GCN | update_timestep | 20 |
| dropout_network | 0.2 | K_epochs | 4 |

- **layer_trans**: Represents the number of MLP layers used to map the initial node embeddings in the GNN encoding part.
- **layer_gnn**: Represents the number of GNN layers used in the GNN encoding part.
- **layer_parsingnet**: Represents the number of layers in the parsing module.
- **gnn_model**: Represents the type of GNN layer used in the GNN encoding.
- **dropout_network**: Represents the proportion of edges randomly dropped in the grouping module.
- **dropout_parsing**: Represents the proportion of edges randomly dropped in the parsing module.
- **link_ignore_self_loop**: Indicates whether to ignore self-loops in the network links.
- **act_final**: Indicates whether to apply activation in the final layer.
- **learning_rate**: Represents the learning rate of the framework.
- **max_episodes**: Represents the number of learning episodes for the framework.
- **update_timestep**: Represents the length of the timestep for exploration within each episode.
- **K_epochs**: Represents the number of policy updates within each episode.

