# OpenReview forum: "A Structure-Aware Framework for Learning Device Placements on Computation Graphs"
_NeurIPS.cc/2024/Conference — NeurIPS 2024 poster_

### Official Review · Reviewer_RMDa · 2024-07-04

**Soundness:** 3
**Presentation:** 3
**Contribution:** 3
**Rating:** 6
**Confidence:** 3

**Summary:**

The paper is about computation graphs, which is an interesting topic. The authors propose a novel framework for the task of device placement, relying on smaller computation graphs extracted from the OpenVINO toolkit using reinforcement learning. The paper is well written and well organized. However, there are several concerns in the current version of the paper that addressing them will increase the quality of this paper.

**Strengths:**

1 Novel ideas.

2 Good writing.

3 Sound experiments.

**Weaknesses:**

1 The author mentioned that concatenating features from different angles would bring benefits. So would concatenating or fusing these features at different dimensional sizes affect their effectiveness?

2 This seems like an interesting research question, but the authors don’t seem to explain the methods used clearly in the main text. I think it is necessary to add an appendix to show the whole framework in the form of a flowchart or pseudocode and explain it in detail.

3 Is a larger baseline model needed for comparison? Large models are popular at present. If it is not necessary, the author can explain the reasoning without adding corresponding experiments.

**Questions:**

As above.

**Limitations:**

The authors provide a discussion of limitations, but it is uncertain whether this is comprehensive.

---

> ### Author Rebuttal · Authors · 2024-08-06
>
> We would like to genuinely thank you for your thoughtful comments and your interest in the selected topic. Please find our response to your comments and suggestions below. We have also updated the manuscript in light of your suggestions towards increasing the quality of our paper.
>
> **1. Concatenating or fusing features at different dimensional sizes.** Concatenating or fusing the initial features at different dimensional sizes indeed affects the effectiveness of the framework. In fact, during our initial experiments, we found that a **well-balanced combination** of feature dimensions results in better model performance and robustness. Therefore, we decided to encode the initial features using equal-sized latent representations. We would like to highlight that the **flexibility** of our framework allows for different feature combination schemes, such as averaging the feature vectors, increasing or decreasing the dimensionality of the feature vectors, or even attending to each feature differently. This is an interesting area for future investigation. We hope that our answer clarifies your question.
>
> **2. Framework explanation and pseudocode.** We would like to thank you for pointing this out and we are pleased you found our research question interesting. Due to the fact that our framework is flexible and can be implemented using a wide range of different components (e.g. using a simple MLP instead of a GCN), the initial goal was to keep the main description of the framework as simple as possible, in an attempt to avoid **information overload** that may confuse and distract the readers. After taking into consideration your suggestion, we have prepared for you a **pseudocode** that describes the entire flow. We will add this pseudocode in the appendix:
>
> **Main Algorithm:** Hierarchical Structure-Aware Device Assignment Graph (HSDAG)
> ---
> 1. Initialize computation graph $G$, coarsened graph $G'$, node features $F$, and maximum iterations $max\_{iterations}$.
> 2. Initialize parameters $\theta$ of Graph Parsing Network (GPN) and Multi-Layer Perceptron (MLP).
> 3. Initialize node assignment matrix $\mathcal{X}$, Adam optimizer, buffer size $x$, and discount factor $\gamma$.
> 4. For $i = 1$ to $max\_{iterations}$ do
>
>    a. Apply GPN: $(C, F_C) \gets GPN(G, F)$, where $C$ are clusters and $F_C$ are cluster features.
>
>    b. Apply MLP: $P' \gets MLP(F_C)$ to get device placement for coarsened graph.
>
>    c. Map $P'$ to original graph $G$ using assignment matrix $\mathcal{X}$.
>
>    d. Deploy $G$ with device placement $P'$ and measure inference latency $l_{P'}(G')$.
>
>    e. Calculate reward: $r_{P'}(G') \gets 1 / l_{P'}(G')$.
>
>    f. Update node embeddings: $\mathbf{Z_v} = \mathbf{Z_v} + \mathbf{Z_{v'}}$ for each node $v$ and its corresponding coarsened node $v'$.
>
>    g. Form a new coarsened graph $G'$ as the new state.
>
>    h. Run a new round of representation and group learning for a new graph $G'$.
>
>    i. Store step information $(P, G', r_{P'}(G'))$ in buffer.
>
>    j. If buffer reaches $x$ steps:
>       - Compute gradient:
>         $\nabla_\theta J(\theta) \approx -\sum_{i=1}^x \nabla_\theta \log p(P | G'; \theta) \cdot \gamma^i \cdot r(P_i, G)$
>       - Update policy parameters $\theta$ using Adam optimizer with the computed gradient.
>       - Clear buffer.
>
>    k. If convergence criteria are met, break the loop.
>
> 5. Return optimal device placement $P_{opt}$.
>
> **Sub-Algorithm: Graph Parsing Network (GPN)**
> ---
> 1. Initialize input graph $G$ with node features $X^{(0)}$ and adjacency matrix $A$.
>
> 2. Initialize learnable parameters $W$ for GNN.
>
> 3. For each iteration:
>
>    a. Perform graph and node encoding:
>       $Z = GNN(X, A) = \sigma(\hat{D}^{-1/2}\hat{A}\hat{D}^{-1/2}X^{(0)}W)$
>       where $\hat{A} = A + I$, $\hat{D_{ii}} = \sum_{j=0} \hat{A}_{ij}$
>
>    b. Calculate edge score matrix:
>       $S_{v,u} = \sigma(\phi(z_v, z_u))$, where $\phi$ is MLP
>
>    c. Apply Graph Parsing Algorithm $\mathcal{A}$:
>       - Stage 1: $\hat{C}^{(k)} \gets DOM(C^{(k)})$
>       - Initialize: $s \gets \emptyset$, $p \gets 0$
>       - Stage 2: While $sum(\hat{C}^{(k)}) \neq 0$ do:
>         * $idx \gets argmax(\hat{C}^{(k)})$
>         * $q \gets |idx|$
>         * $(l, idx') \gets EXP(idx, \hat{C}^{(k)})$
>         * $idx \gets union(idx, idx')$
>         * While $|idx| = q$:
>           + $s \gets union(s, \{(i,p)|i \in l\})$
>           + $\hat{C}^{(k)}_{i,j} \gets 0, \forall (i,j) \in idx$
>           + $p \gets p + 1$
>       - Stage 3: $S^{(k)} \gets GEN(s)$
>
>    d. Create node assignment matrix:
>       $\mathcal{X} = \mathcal{A}(\mathcal{E})$
>
>    e. Define adjacency matrix for pooled graph:
>       $A' = \mathcal{X}^T \cdot A \cdot \mathcal{X}$
>
> 4. Return pooled graph $G'$ and node assignment matrix $\mathcal{X}$
>
>
> **3. LLM comparison.** Thank you for the question. Experiments on LLMs would be beneficial to have although not strictly needed for demonstrating the effectiveness of our method. Given that our method is not strictly dependent on the model architecture, and that we showed its effectiveness on a diverse set of architectures, one of which was **transformer-based** (the foundation of LLMs), we do believe that our method would be scalable and effective for LLM as well. **Reviewer wsma** made a quite similar comment. We kindly ask you to have a look at this response as well, as it provides additional context.
>
> **Limitations.** Thank you for your comment. We kept the discussion of limitations section within the required space limits. We will elaborate more on this section and provide a revised version in the appendix.

---

> > ### Comment · Reviewer_RMDa · 2024-08-12
> >
> > Thanks for the author's reply, I have no doubts anymore.

---

> > > ### Author Response · Authors · 2024-08-13
> > > **Thank you!**
> > >
> > > Thank you for carefully reading our response. We are glad that our response addressed all of your concerns.

---

### Official Review · Reviewer_PWmk · 2024-07-11

**Soundness:** 4
**Presentation:** 4
**Contribution:** 4
**Rating:** 7
**Confidence:** 3

**Summary:**

The authors have developed a model that automatically places neural network operations on the optimal devices (CPU and GPU) to accelerate model training. They optimize device placement using reinforcement learning and use execution time as the reward. The authors propose using the graph structure as information to choose the appropriate device for each operation. They evaluate their method based on execution time gains compared to several baselines and demonstrate significant improvements on three architectures.

**Strengths:**

- The paper is very well written, and all the steps of the model are well detailed, making it easy to understand.
- The experiments reflect the quality of their method compared to the state-of-the-art.
- The ablation studies show the impact of each component of the model.

**Weaknesses:**

/

**Questions:**

I did not understand how the computation graph of the neural network was retrieved. Could you please provide more details?
Is there any impact of the chosen GNN architecture, such as GCN or GAT, for example?

**Limitations:**

I see no ethical limitations; the checklist is well completed, and clear justifications are provided.

---

> ### Author Rebuttal · Authors · 2024-08-04
>
> We would like to thank you for the kind words and we are glad that you liked our paper. We provide clarifications on details that may have been unclear.
>
> **Computation graph.** To convert a neural network model to an OpenVINO computation graph, we first design the model using a deep learning framework like PyTorch. Next, we convert and save the model to OpenVINO's Intermediate Representation (IR), which consists of an XML file representing the model's structure. By reading this XML file, we extract the necessary information to understand the model's layers and their connections. Finally, we use this information to build a directed acyclic graph that represents the computation process of the neural network, ensuring each layer is properly mapped and connected.
>
> **Impact of the chosen GNN architecture.** We would like to thank the reviewer for posing this interesting question. There is **no major impact** in choosing a different GNN architecture, such as GCN, GAT, GIN or GraphSAGE. We selected GCN since it is the **de-facto GNN** layer every baseline model uses and the one that was proposed by the paper on graph parsing networks [1]. To verify whether a GNN architecture critically affects the performance of our framework, we run more experiments with different GNN layers. We provide the results in the table below (execution time in seconds):
>
> |           | Inception-V3 | ResNet  | BERT   |
> |-----------|-------------|---------|--------|
> | GCN       | 0.0105      | 0.00766 | 0.00267|
> | GAT       | 0.0109      | 0.00772 | 0.00272|
> | GIN       | 0.0104      | 0.00770 | 0.00269|
> | GraphSAGE | 0.0102      | 0.00767 | 0.00266|
>
>
> **References**
> 1. Song, Yunchong, et al. **"Graph Parsing Networks."** arXiv preprint arXiv:2402.14393 (2024).

---

> > ### Comment · Reviewer_PWmk · 2024-08-10
> >
> > Thank you for your response and your clarifications. I'll keep my score as it.

---

> > > ### Author Response · Authors · 2024-08-10
> > > **Appreciation for your feedback!**
> > >
> > > Thank you for your thorough review and feedback. We appreciate your positive evaluation and are grateful for your support of our work.

---

### Official Review · Reviewer_orCV · 2024-07-12

**Soundness:** 2
**Presentation:** 2
**Contribution:** 2
**Rating:** 4
**Confidence:** 3

**Summary:**

This paper introduces an end-to-end framework utilizing policy optimization for device placement during the inference phase of deep learning models. The framework consists of multiple components, including computation graph construction, graph coarsening, node representation learning, and policy optimization. Evaluations using benchmark models such as Inception-V2, ResNet, and BERT demonstrate improved CPU execution speedup compared to several device placement methods.

**Strengths:**

+This paper presents a nice overview figure of the proposed framework with a detailed description of each step in the framework.

+The paper has provided code and implementation details.

**Weaknesses:**

-After reading the paper, I do not think the motivation of the designed framework is strong. It is not clear to me what the rationale of each component and the choice of this policy optimization method is.

-Some suggestions on paper presentation: In the abstract, it would be helpful to introduce the problem of device placement, and the concepts of “computation graphs” and “heuristic methods” in this problem. It is not clear to me why ignoring the topological features of computation graphs is an issue.

-Some redundancy can be reduced. For instance, in definition 2.2, “a device p in D, where p in {1,2,…,|D|}.”
-The problem setup is confusing. There are also some inconsistent notations between the text and the equations. It would be nice to have some descriptions of the device placement process.

-It seems that graph structural features do not help that much in the speedup from Table 3.

-It would be useful to include the complexity comparison of the proposed method with other baseline methods.

**Questions:**

1)	In the abstract, the last sentence is confusing: “the suggested placements improve the inference speed for the benchmark models by up to 58.2% over CPU execution and by up to 60.24% compared to other commonly used baselines”.  What is the improvement of 60.24% measured on? Is this accuracy or CPU execution?

2)	Some notations in Definition 2.2 are not clear. What is the index n in a placement P? Is this equivalent to the number of nodes |V| or to the number of devices |D|?

3)	In the problem setup, is there any order when assigning operations to a device? Why do we need to use policy learning?

4)	What does HSDAG stand for?

**Limitations:**

Some limitations are discussed in the conclusion section.

---

> ### Author Rebuttal · Authors · 2024-08-06
>
> We thank the reviewer for the insightful comments. We hope our response addresses your concerns and highlights the framework’s motivation and rationale.
>
> **TL;DR:** Our high-level motivation is to address the ever-growing capacity requirements for efficient inference on heterogeneous devices. Please refer to the first two paragraphs of similar work **“Placeto: Learning Generalizable Device Placement Algorithms for Distributed Machine Learning”** where the motivation and the choice of RL for this problem are nicely consolidated. Admittedly, our motivation is not as well-presented. We will make some minor restructuring and make it more clear.
>
> **In the following text, we have prepared for you a detailed explanation of our rationale, which we hope addresses your concerns. We invite you to read it and please let us know if you think we should briefly mention/highlight any of these in the introduction.**
>
> **Limitations, motivation, rationale:**
> 1. **Limitation:** Not capturing directed interactions among nodes. **Why is this a limitation?** The model runs poorly in graph coarsening. **Solution:** We use local/global structural features. Our ablation study verifies that such features increase the model’s performance.
> 2. **Limitation:** Relying only on heuristics or simple methods (e.g. k-means) for graph partitioning. **Why is this a limitation?** Such methods require hyperparameter tuning for their components to obtain good performance. This ad-hoc presetting of the group number inhibits the exploration, generalization and learning process of a framework. **Solution:** We learn how to partition a given graph into an unspecified number of groups. Both the number of node groups and the pooling algorithm are end-to-end learnable parameters. Our framework facilitates “personalized” partitioning: the number of node groups is dynamically decided depending on the scale of the computation graph and a more sophisticated algorithm for pooling is learned.
> 3. **Limitation:** Following either grouper- or encoder-placer architecture. **Why is this a limitation?** In both cases, end-to-end training is not allowed as the encoding and coarsening steps run sequentially and individually. Thus, their components are not trained based on the optimal device placements. **Solution:** We fuse encoder- and grouper-placer techniques and effectively combine their strengths to jointly optimize end-to-end representation learning and graph coarsening phases interdependently.
> 4. **Limitation:** Not supporting simultaneously training all steps in an end-to-end fashion. **Why is this a limitation?** Supporting end-to-end training is necessary to ensure each step is tailored to the final device placement. **Solution:** We carefully designed our framework to enable end-to-end training, where all components are trained simultaneously.
> 5. **Limitation: Ignoring topological features (also an answer to weakness 4)**. **Why is this a limitation?** Topological features allow frameworks to model the order of nodes. **Solution:** Our framework learns how to better partition a computation graph, leading to better performance. Our ablation study shows the benefits of topological features.
>
> **Introducing concepts, redundancy, inconsistent notations.** We have improved the abstract by introducing the device placement problem and mentioning the concepts of computation graphs and heuristic methods. We have removed redundancies and improved the notation.
>
> **Graph structural features.** The varying impact of graph structural features over different models correlates with the complexity of computation graphs. Inception-V3 has a complex computation graph and an architecture with multiple parallel branches leading to higher values in all metrics (see table below). For such complex graphs, GNN approaches struggle to extract all essential features. So, adding structural features enhances their ability to model important graph characteristics. Conversely, simpler graphs allow GNNs to extract relevant features without external support. So, graph structural features have less of an impact on their performance.
>
> |Metric|BERT|Inception-V3|ResNet|
> |--------|------|--------------|--------|
> |Graph Diameter|34|**49**|32|
> |Average Fractal Dimension|0.00412|**0.00427**|0.00392|
> |Degree of Heterogeneity|1.247|**1.367**|1.268|
>
> **Framework’s Complexity.** Please take a look at a similar response to **Reviewer wsma**.
>
> **Answers to questions:**
>
> **Q1.** Both values refer to execution runtime. As CPU-only execution is one of the most important baselines, we highlight our performance against it. 60.24% is the improvement of the execution speed against Placeto. We have slightly rephrased to avoid confusion.
>
> **Q2.** The index n in a placement $P$ is the number of nodes $|V|$. We included the missing notation.
>
> **Q3 Assigning order and policy learning.** We understand your concern about using policy methods for this problem as the use of RL might not be directly suggested. We invite you to have a quick look at similar studies below:
> 1. Figure 2 and Section 2.1 - “MDP Formulation” of **“Placeto: Learning Generalizable Device Placement Algorithms for Distributed Machine Learning”**: The device placement is modeled as an iterative process where the algorithm inputs a current placement, picks a node and outputs a device for that node.
> 2. Figure 1 in **“Accelerated Device Placement Optimization with Contrastive Learning”**: The state-space/observation is the entire computation graph and the input to the policy algorithm. The output is an assignment that maps each node to a device. Our problem formulation is similar to this one.
> In both cases, the order of assigning nodes to devices does not matter. This is not due to the RL problem formulation but rather due to the nature of the problem. The reward only depends on the final device placement.
>
> **Q4 HSDAG** = Hierarchical Structure-Aware Device Assignment Graph

---

> > ### Comment · Reviewer_orCV · 2024-08-12
> >
> > Thanks for addressing my concerns. I just wish the context information is included in the paper as a paper should be self-contained and the background knowledge is important in this research topic.

---

> > > ### Author Response · Authors · 2024-08-12
> > >
> > > Thank you for your feedback. We are glad that our response addressed your concerns. We will make sure all the context information is included in the final version of the paper. We welcome any further discussions.

---

### Official Review · Reviewer_wsma · 2024-07-19

**Soundness:** 4
**Presentation:** 3
**Contribution:** 3
**Rating:** 6
**Confidence:** 4

**Summary:**

The paper introduces a novel framework for device placement that leverages smaller computation graphs extracted from the OpenVINO toolkit using reinforcement learning. This framework bridges the gap between encoder-placer and grouper-placer techniques by incorporating graph coarsening, node representation learning, and policy optimization. It supports end-to-end training and considers the directed and acyclic nature of computation graphs. The framework also includes a model variant inspired by graph parsing networks and complex network analysis, enabling joint learning of graph representation and personalized graph partitioning. Experimental results demonstrate significant improvements in inference speed for benchmark models like Inception-V3, ResNet, and BERT.

**Strengths:**

* The framework effectively combines graph coarsening, node representation learning, and policy optimization, allowing for a comprehensive approach to device placement that captures both local and global structural features.
* The ability to train all components of the framework in an end-to-end fashion ensures that the model can learn optimal device placements efficiently, improving overall performance.
* The proposed framework demonstrates substantial improvements in inference speed, achieving up to 58.2% over CPU execution and up to 60.24% compared to other baselines, highlighting its effectiveness and robustness.

**Weaknesses:**

* Could the strategy that uses the execution time only of the suggested device placements as a reward to train the proposed framework hinder the generalization of it to other scenarios? Are there any other rewards that the authors attempt to incorporate into their framework?
* Besides the inference latency, could the authors provide the results of different models on the standard benchmarks to reflect that HSDAG could not only accelerate the deployment but also not affect the performance of downstream tasks?
* The complexity of the proposed framework and comparison of the running time for getting the final allocation strategy with baselines can further strengthen the manuscript.
* While the authors elicit the challenges from the perspective of foundation models, could the authors showcase the potential ability of HSDAG in handling foundation models such as LLMs?
* Minor: The title should follow the Research Paper Title Capitalization Rules. $n=|V
|$ is the number of nodes that should be noted in Definition 2.2.

**Questions:**

Please kindly refer to the Weaknesses.

**Limitations:**

The limitations are discussed after the conclusion.

Minor: There is a typo in the limitation section: "We attempted to obtain \underline{the he} source codes for the baseline methods".

---

> ### Author Rebuttal · Authors · 2024-08-06
>
> Thank you very much for recognizing the novelty of our framework, and highlighting its effectiveness and robustness. Below we provide point-to-point responses to your questions/concerns. We hope that our response addresses your concerns.
>
> **Execution time as a reward.**  Thank you for this remark. Most of the existing work on device placement (including our baseline models) uses the execution time as a reward and as the most important evaluation metric since, ultimately, the goal is to speed up the inference time [1,2]. Hence, in this paper, we follow the same protocol. The experimental results (Table 2, page 8)  from both our approach and the baseline models demonstrate that the generalization **is not compromised** by following a strategy to minimize the execution time. Generalization would potentially come into play in the case of a **reward model**, which would make sense for this problem given the difficulty in measuring the reward. Another option for this problem would be to use **incremental rewards** like those mentioned in Section 2.1 of **“Placeto: Learning generalizable device placement algorithms for distributed machine learning”** which, however, do not quite fit our RL problem formulation (but which, interestingly enough, would benefit from a reward model). Exploring the reward model direction, potentially incorporating incremental rewards (the combination of which would allow us to leverage the low variance and faster convergence associated with less sparse rewards) and studying how it impacts generalization, is a very interesting direction for future work. We will briefly mention this in the conclusion.
>
> **Model performance in downstream tasks.** Thanks for your comment and for pointing this out. Theoretically, since the algorithm itself does not change, the performance of the model in downstream tasks will remain the same. For further verification, we conducted experiments using the **Inception-V3**, **ResNet** and **BERT** models and provide the results below:
>
> 1. **Inception-V3:**  We performed image classification inference on images depicting Samoyed dogs. All the parameters are directly derived from the torchvision pre-trained model. We did not change any configuration on the data type of the model. The classification accuracy of Inception-V3 using the best device placement is **82.77%**. For the GPU-only experiments the classification accuracy is **82.72%**. For the CPU-only experiments the classification accuracy is **82.33%**.
>
> 2. **ResNet:** Similarly, we performed image classification inference using the ResNet model on the same dataset. The classification accuracy with the best device placement is **45.37%**. For the GPU-only experiments the classification accuracy is **45.37%**. For the CPU-only experiments the classification accuracy is **45.44%**.
>
> 3. **BERT:** We evaluated the performance of the BERT model using the output embeddings from the different device placements. We calculated their mean squared error, cosine similarity and Euclidean distance (MSE: the lower the better, cosine similarity: the higher the better, euclidean distance: the lower the better):
>
> | Comparison|Mean Squared Error (MSE)|Cosine Similarity|Euclidean Distance (L2 norm)|
> |------------------|---------------------------|-----------------------|------------------------------|
> |CPU vs GPU| 3.04970071738353e-05|0.9999468922615051|0.4328667223453522|
> |**CPU vs HSDAG**|**6.819997793172661e-07**|**0.9999988079071045**|**0.06473180651664734**|
> |GPU vs HSDAG|3.174534140271135e-05|0.9999447464942932|0.44163718819618225|
>
> **The conducted experiments demonstrate that HSDAG does not affect the performance of the model in the downstream tasks. All models have similar performance regardless of the running device (e.g. CPU, GPU or heterogeneous device).**
>
> **Framework’s complexity.**  Thank you very much for the suggestion on calculating the complexity of the proposed methods. We provide a table with the running time complexity in seconds of our method and the baseline methods. As it is shown, **HSDAG** is **significantly faster** in all cases. We agree that these additional experiments do indeed **strengthen our manuscript** and we will make sure to include them in the appendix.
>
> |             | Inception-V3 | ResNet | BERT  |
> |-------------|--------------|--------|-------|
> | Placeto| 2808s| 1162s  | 4512s |
> | RNN-based| 3706s| 1212s  | OOM|
> | **HSDAG**| **2454s**| **1047s**  | **2765s** |
>
>
> **HSDAG handles LLMs.** We indeed elicit the challenges from the perspective of foundation models. We believe that the ever-rising capacity requirements and cost associated with LLMs will make the problem of device placement even more important. Since: 1) our method is not strongly dependent on the architecture or scale of the underlying model, 2) we showcased the effectiveness in diverse architectures and 3) one of those architectures was a **transformer-based** model (BERT) which is the foundation of LLMs, we do believe that HSDAG is extensible and applicable to LLMs. Nonetheless, LLMs pose their own unique challenges and without a strong empirical evaluation on diverse LLM architectures (which would be extensive enough to produce another manuscript), we did not want to make any claim about the extensibility and scalability of our method. We will add 2 short sentences in the conclusion to highlight this future direction and as well as its unique challenges.
>
> **Typos, capitalization rules and missing notation.** We will update and fix it accordingly. Thank you.
>
> **References**
> 1. Mirhoseini, Azalia, et al. **"Device placement optimization with reinforcement learning."** International conference on machine learning. PMLR, 2017.
> 2. Addanki, Ravichandra, et al. **"Placeto: Learning generalizable device placement algorithms for distributed machine learning."** arXiv preprint arXiv:1906.08879 (2019).

---

> > ### Comment · Reviewer_wsma · 2024-08-09
> >
> > Thank you for the rebuttal. I have also read the reviews from other reviewers as well as the corresponding reply. I have no further questions and I choose to increase my current rating accordingly.

---

> > > ### Author Response · Authors · 2024-08-09
> > > **Thank you for raising the score!**
> > >
> > > Thank you for carefully reviewing all the reviews and replies, as well as for your timely response. We are glad to see that your concerns have been addressed and thank you for the updated score.

---

### Author Rebuttal · Authors · 2024-08-06

We thank all the reviewers for their careful reading and thoughtful comments and suggestions on our paper. We find it encouraging that reviewers have found our work **interesting**, **novel** and **well-organized**! The suggestions have led us to further improve the clarity of the manuscript, improve minor issues as well as add some more technical details. We have addressed the individual reviewer’s comments below. Here, we **summarize** the proposed changes, which we hope are within the extent of **allowed modifications** and would not fundamentally alter the main method and results.

**Changes:**

* Additional experiments: We will add the results of the auxiliary experiments conducted during the rebuttal period to the Appendix of the paper. These experiments:
    1. Demonstrate that the performance of downstream tasks is indeed **not affected** by our framework (**Reviewer wsma**);
    2. Show the **complexity** of the framework (**Reviewer orCV**, **Reviewer wsma**);
    3. Demonstrate the **importance** of the graph structural features (**Reviewer orCV**).
* Fixing notation issues, typos, and briefly introducing the problem definition in the abstract (**Reviewer orCV**, **Reviewer wsma**).
* Slightly restructuring the introduction to make the **high-level motivation** more clear (**Reviewer orCV**).
* Summarize our entire flow in a **pseudocode** block to be added in the appendix (**Reviewer RMDa**).
* Add a few future research directions and limitations in the conclusion and appendix, respectively (**Reviewer RMDa**, **Reviewer wsma**).

---

### Decision · Program_Chairs · 2024-09-25

**Decision:**

Accept (poster)

**Comment:**

This paper presents a novel structure-aware framework using reinforcement learning to optimize device placement in computation graphs, showing substantial improvements in inference speed for deep learning models. The approach effectively incorporates graph structural features, providing a robust solution that bridges existing gaps in device placement techniques. While initial concerns were raised regarding the clarity of motivation and theoretical contributions, the authors' detailed rebuttal successfully addressed these points, demonstrating the framework’s practical relevance and strong potential for real-world applications.